# Immunomics-guided biomarker discovery for human liver fluke infection and infection-associated cholangiocarcinoma

Lakkhana Sadaow[1,10], Rutchanee Rodpai[1,10], Michael J. Smout [2,10], Rie Nakajima[3,10], Patcharaporn Boonroumkaew[1], Javier Sotillo [4], Bemnet A. Tedla[2], Vor Luvira[5], Amnat Kitkhuandee[5], Krisada Paonariang[5], Wattana Sukeepaisarnjaroen[6], Hiroshi Yamasaki[7], Sutas Suttiprapa[8], Thewarach Laha[1], Banchob Sripa[8], Rafael de Assis[3], Aarti Jain[3], Wannaporn Ittiprasert[9], Victoria H. Mann[9], Yide Wong[2], Philip L. Felgner [3,11], Wanchai Maleewong[1,11], Paul J. Brindley [9,11] ✉, Alex Loukas [2,11] ✉ & Pewpan M. Intapan[1,11]

Sensitive diagnostics are needed to improve management and surveillance of opisthorchiasis and opisthorchiasis-associated cholangiocarcinoma (CCA) throughout East Asia. Herein we generate and screen an *Opisthorchis viverrini* recombinant secreted proteome to identity antibody biomarkers of liver fluke infection and CCA with sera from study participants in endemic populations and evaluate their utility as point-of-care immunochromatographic tests (PoC-ICTs). We incorporate two of the most promising antigens from the proteome array screen, P1 and P9, into PoC-ICTs to further validate their diagnostic performance. The P9-IgG4 PoC-ICT is superior amongst the single recombinant antigen tests for diagnosing fluke infection as well as fluke-induced CCA, and out-performs parasite crude extract-IgG ICTs. Here we identify two biomarkers of *O. viverrini* infection and infection-associated CCA that could form the basis of novel antibody serodiagnostic tests for human liver fluke infection and associated cancer.

Food-borne trematodiases affect more than 50 million people globally[1]. Of these trematode flukes, arguably the most important are the carcinogenic liver flukes, *Opisthorchis viverrini*, *Opisthorchis felineus* and *Clonorchis sinensis*. One hundred million residents of the lower Mekong River drainage region are at risk of infection with *O. viverrini*, with ~10 million infected in the north-eastern region of Thailand, two million in the Lao People's Democratic Republic and many more infections in southern Vietnam, Cambodia and Myanmar (reviewed in[2]).

*O. viverrini* infection is recognised by the WHO as a group 1 biological carcinogen because of its cogent link with cholangiocarcinoma

[1]Department of Parasitology, Faculty of Medicine, and Mekong Health Science Research Institute, Khon Kaen University, Khon Kaen, Thailand. [2]Australian Institute of Tropical Health and Medicine, James Cook University, Cairns, QLD, Australia. [3]Vaccine R&D Center, Department of Physiology and Biophysics, University of California Irvine, Irvine, CA, USA. [4]Parasitology Reference and Research Laboratory, Centro Nacional de Microbiologia, Instituto de Salud Carlos III, Majadahonda, Madrid, Spain. [5]Department of Surgery, Faculty of Medicine, Khon Kaen University, Khon Kaen, Thailand. [6]Department of Medicine, Faculty of Medicine, Khon Kaen University, Khon Kaen, Thailand. [7]Department of Parasitology, National Institute of Infectious Diseases, Tokyo, Japan. [8]Tropical Disease Research Center, Department of Tropical Medicine, Faculty of Medicine, Khon Kaen University, Khon Kaen, Thailand. [9]Department of Microbiology, Immunology and Tropical Medicine, School of Medicine and Health Sciences, George Washington University, Washington, DC, USA. [10]These authors contributed equally: Lakkhana Sadaow, Rutchanee Rodpai, Michael J. Smout, Rie Nakajima. [11]These authors jointly supervised this work: Philip L. Felgner, Wanchai Maleewong, Paul J. Brindley, Alex Loukas, Pewpan M. Intapan. ✉e-mail: pbrindley@gwu.edu; alex.loukas@jcu.edu.au

(CCA), cancer of the biliary tract[3]. The incidence of CCA associated with liver fluke infection varies by geographical region and other risk factors but has exceeded 100 per 100,000 in men and 40 per 100,000 in women in hotspots in northeast (NE) Thailand[4]. Infection with *O. viverrini* is acquired by consumption of raw, undercooked or fermented freshwater fish which harbour the larval stage of the parasite, termed the metacercaria. Ingested metacercariae excyst in the stomach and juvenile flukes migrate up the ampulla of Vater to the biliary tract. The adult flukes mature within the bile ducts, and feed on the biliary epithelium. Adult flukes are very long lived (more than 10 years), so infected persons experience chronic cycles of wounding and healing of the bile duct tissue over many decades. Fluke infection in NE Thailand and Laos is frequently coupled with elevated levels of dietary nitrosamine compounds (from fermented fish and other foods), culminating frequently in the emergence of carcinogenesis[2].

Sustainable control of liver fluke infection, whether through public health and education approaches, including mass drug administration[5] or ultimately through vaccination[6], requires specific and sensitive diagnostic tools that are readily deployable in the field and easy to use. Methods to detect infection need to be appropriately sensitive and rapid to diagnose new cases, assess effectiveness of elimination measures and be applicable to large-scale disease surveillance. Indeed, the WHO recently highlighted the low sensitivity of current diagnostic methods as a barrier to controlling liver fluke infection[7]. The most widely used method for diagnosing infection is microscopy-based formalin ether concentration technique (FECT)[8]. FECT for diagnosing opisthorchiasis exhibits relatively poor sensitivity in areas of low transmission, limiting its value as a diagnostic tool. Antigen-detection and antibody-detection tests for opisthorchiasis have been reported, but few of them utilize reagents that have been fully characterised. For example, capture assays to detect liver fluke antigen in urine and feces display improved sensitivity compared to FECT, but they rely on monoclonal antibodies (mAb) to capture antigens that have been poorly defined, and likely target carbohydrate epitopes[8,9]. Recently, an immuno-chromatographic test (ICT) using a mAb was developed to detect *O. viverrini* antigen in urine[10], although its antigenic target also remains unknown. Antibody-based tests using crude somatic or excretory/secretory (ES) antigen preparations are used in the field but are less specific than FECT[11]. Moreover, diagnostic tests that rely on parasite extracts suffer from an absence of quality control and rely on replenishment of *O. viverrini* parasites from the field because the entire life cycle of the parasite is difficult to maintain in the laboratory.

A handful of serodiagnostic recombinant antigens have been reported for liver fluke infection but there has not been a systematic and unbiased screen to date. We and others have previously used proteome array technologies to study the humoral response to other helminth infections, probing the arrays with infection sera to aid in genome-wide vaccine antigen and serological marker discovery[12,13].

Herein, we have leveraged the *O. viverrini* surface and secreted proteomes to produce the first *O. viverrini* protein microarray. In an integrated approach to identify diagnostic biomarkers for opisthorchiasis and liver fluke-induced CCA, we probed the secreted proteome (secretome) array with sera from individuals from geographically distant *O. viverrini*-endemic regions in Thailand and PDR Lao to determine targets of IgG responses. Subsequently, candidate proteins were sprayed onto pilot ICTs and seroreactivity validated for the best performing antigens. Two antigens were particularly immunoreactive during both infection and CCA, and these ICTs now warrant in-depth assessment for their predictive value in the field.

## Results

### Characterisation of the *O. viverrini* secretome using mass spectrometry

A proteomic analysis of the *O. viverrini* secretome, including soluble ES products as well as a re-assessment of previously published EV

proteomics data was performed to select the most promising protein targets for the array. A total of 825 proteins were identified with ≥2 peptides between all datasets – 279, 527 and 698 in the ES, exosome-like vesicles (ELVs) and microvesicles (MVs) of *O. viverrini*, respectively (Fig. 1, Supplementary Data 1, Supplementary Fig. 1). Of these proteins, 163 were common to all 3 samples, and 45 were present in just 2 of the 3 samples. Proteins were manually curated, and 245 were advanced to printing based on their abundance, presence in multiple samples, similarity to known diagnostic antigens in liver fluke and other related fluke infections, and completeness of the ORF. Select purified recombinant proteins from *O. viverrini* also were also printed on the array due to their proven roles in host-parasite interactions[14–16]. Furthermore, two *C. sinensis* proteins were also selected based on their previously published diagnostic capacity[17], making a total of 249 proteins selected for the array. Since 18 of these proteins were large (of high molecular mass), they were fragmented into smaller polypeptides before being printed onto the array, leading to a total of 278 printed proteins or protein fragments.

### *O. viverrini* sero-reactome determined using immunomics

Serum IgG subclass responses to numerous arrayed antigens were elevated in the infected population when compared with the non-infected samples from both Thailand (*O. viverrini* endemic site but FECT-negative) and USA (non-endemic site). For statistical comparisons, we compared just *O. viverrini* FECT positive and non-endemic uninfected subjects from the USA because uninfected endemic subjects may have been previously and/or continuously exposed but remained FECT-negative. Thirty-six (36) antigens were the targets of significantly elevated IgG1 or IgG4 responses in the sera of individuals who were infected, and 20 were targets of both significantly elevated IgG1 and IgG4 (Fig. 2a) responses.

Next, we probed the array with sera from human subjects infected with the related liver fluke, *C. sinensis* to observe the degree of immunologic cross-reactivity to arrayed *O. viverrini* antigens. Some antigens were recognized exclusively by *O. viverrini* infected individuals (e.g., P4, P8 and P9 IgG1 and IgG4) whereas others were recognized most strongly by *C. sinensis* infection sera (e.g., P3 IgG4), probably due to the higher average infection intensity in the *C. sinensis* infected subjects compared to the *O. viverrini* infected subjects (Fig. 2b). We also probed the array with sera from 50 cases of *O. viverrini*-induced CCA to determine whether IgG antibodies targeting fluke antigens could perform as a cancer biomarker. IgG1 targeting P1, P2 and P3 was the dominant isotype in this cohort of cancer patients when probing the array (Fig. 2b).

P1, P5, P6, P8 and P9 were produced in recombinant form using *E. coli* and purified for incorporation into ICTs to validate their seroreactivity from the protein arrays. We routinely found that anti-IgG1 secondary antibodies were sub-optimal for serodiagnosis in ICTs whereas anti-IgG antibodies (where IgG1 is the dominant subclass) yielded more specific and sensitive results. As such, IgG and IgG4 ICTs were probed with the same serum samples from the control, *C. sinensis*-infected, and CCA cohorts, however we also used an additional cohort of 114 *O. viverrini*-infected cases (Lao PDR and Thailand) with low, medium, and high intensity infections (Fig. 3). ICT and proteome array reactivities were not always aligned. For example, proteome array screening showed that P1 was not able to significantly distinguish between *O. viverrini* infected and controls for IgG or IgG4 antibodies, whereas P1-IgG-ICT was the best protein at distinguishing infected from uninfected subjects with IgG (Supplementary Fig. 2). In contrast, IgG and IgG4 against P8 was able to significantly distinguish between infected and uninfected subjects on the array (Fig. 2b) but was less reactive in ICT form (Fig. 3).

The best performing antigens for diagnosing opisthorchiasis using the IgG-ICT and IgG4-ICT were P1 and P9, respectively (Fig. 4a, b), and both proteins were detected in all three secretome fractions

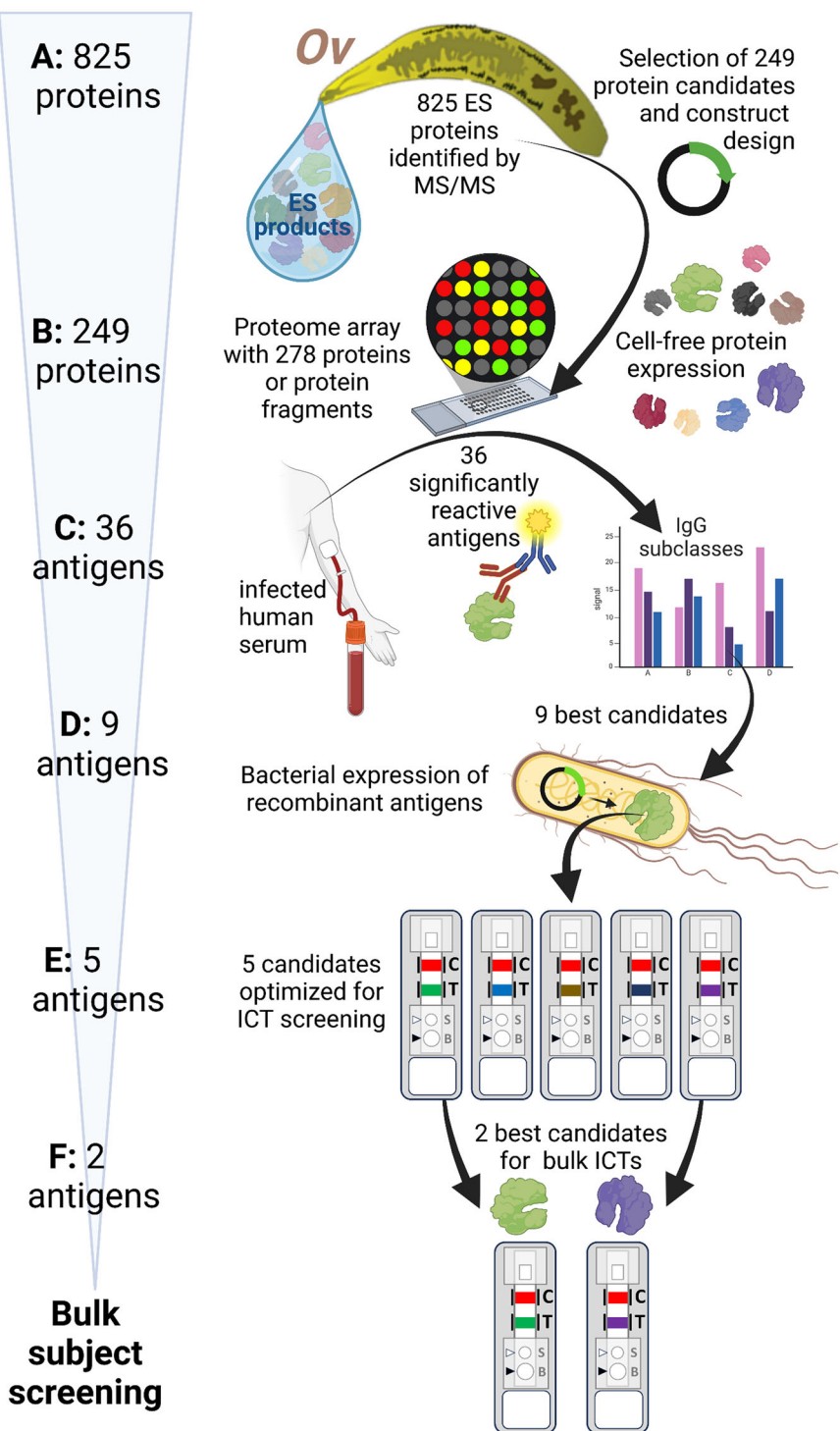

**Fig. 1 | Summary of immunomics approach to antigen discovery and validation using point-of-care immunochromatographic tests (PoC-ICT) for diagnosis of opisthorchiasis and opisthorchiasis-associated cholangiocarcinoma.** We used an integrated approach to identify biomarkers of *Opisthorchis viverrini* (Ov- curved yellow shape) infection and associated cholangiocarcinoma. **A** 825 total excreted/secreted (ES) proteins (water drop) were considered for investigation. **B** 249 proteins were selected for cell-free expression and 245 were successfully expressed. Large proteins were broken up into two ORFs for cell-free expression, resulting in a total of 278 proteins or protein fragments being printed onto a proteome microarray. Four *O. viverrini* ES proteins identified in previous studies and expressed in *E. coli* and purified were also printed onto the array as controls. **C** The proteome array was screened with sera from *O. viverrini* infected subjects, and 36 antigens were the targets of significantly ($p < 0.05$) elevated IgG or IgG4 responses relative to healthy (uninfected) controls. **D** The top nine candidates were selected for bacterial expression. **E** Five candidates were successfully expressed, purified and printed onto pilot immunochromatographic tests (ICTs). **F** The two best performing candidates were incorporated into ICTs for serological validation. Image created in Biorender. Smout, M. (2025) https://BioRender.com/2i9y022.

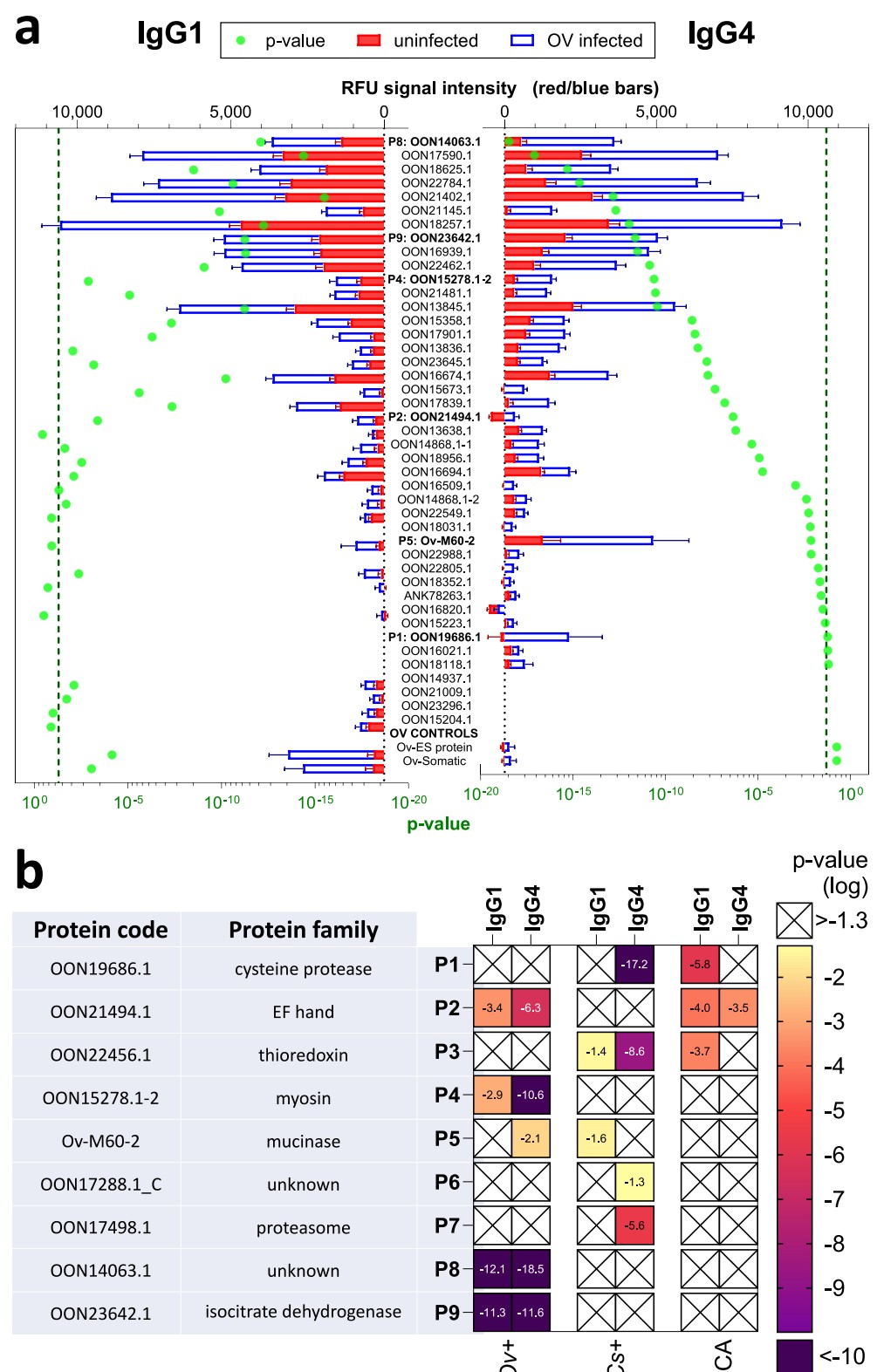

(Supplementary Data 2). We further analysed the antibody responses to these two proteins, stratifying subjects based on infection intensity. Significant positive correlations were detected between P1-IgG reactivity and EPG (r = 0.432, p < 0.001) and between P9-IgG4 reactivity and EPG (r = 0.450, p < 0.001), where increasing EPG correlated with increasing ICT score (Fig. 4c, d). We further validated the P1-IgG and P9-IgG4 ICTs with a different cohort of sera from 50 subjects from Khon

Kaen province who were determined to be FECT-positive for *O. viverrini* infection but for whom quantitative egg counts were not available (subjects were scored simply as positive or negative). Similar positivity rates were obtained as those seen with quantified egg counts, further validating our ICTs (Supplementary Fig. 3). P9-IgG and P1-IgG4 ICTs were less effective at diagnosing infections (Supplementary Fig. 4) and significant correlations with EPG were not apparent (not shown).

**Fig. 2 | IgG1 and IgG4 antibody responses to *Opisthorchis viverrini* recombinant proteins printed on a proteome microarray.** Top 40 protein hits ranked by t-test significance comparing IgG1 and IgG4 (**a**) responses from 100 negative subjects (50 *O. viverrini* fecal egg count test (FECT)-negative subjects living in an endemic region of Thailand and 50 non-endemic negative subjects residing in the US, red bars) and 50 *O. viverrini* FECT-positive subjects (Thailand, blue/white bars). RFU: relative fluorescence units. Bars indicate standard error of the mean (SEM). *p*-value comparing infected and uninfected negative control responses shown as green dots with green dashed line marking the *p* < 0.05 threshold. Protein accession numbers are provided on the y axis, and fluke extracts (OvES, excretory/secretory antigen; OvSo, somatic lysate antigen) were printed as controls. Immunoreactive

proteins selected from the proteome array screen (**b**) for the serodiagnosis of opisthorchiasis and opisthorchiasis-associated cholangiocarcinoma (CCA). Protein accession codes and biological functions (families) are listed, and for ease of nomenclature designated as P1-P9. The heatmap reflects the degree of statistical significance for IgG1 or IgG4 antibody levels (relative fluorescence units) in *O. viverrini* FECT positive subjects (Ov + ), *Clonorchis sinensis* FECT positive subjects (Cs + ) or CCA patients compared to healthy controls (both endemic FECT-negative and US donors). Significance values < 0.05 are denoted by colour, scaling from least significant (yellow) to most significant (purple). "X" denotes *p* > 0.05 or where the healthy control signal was higher than the disease groups.

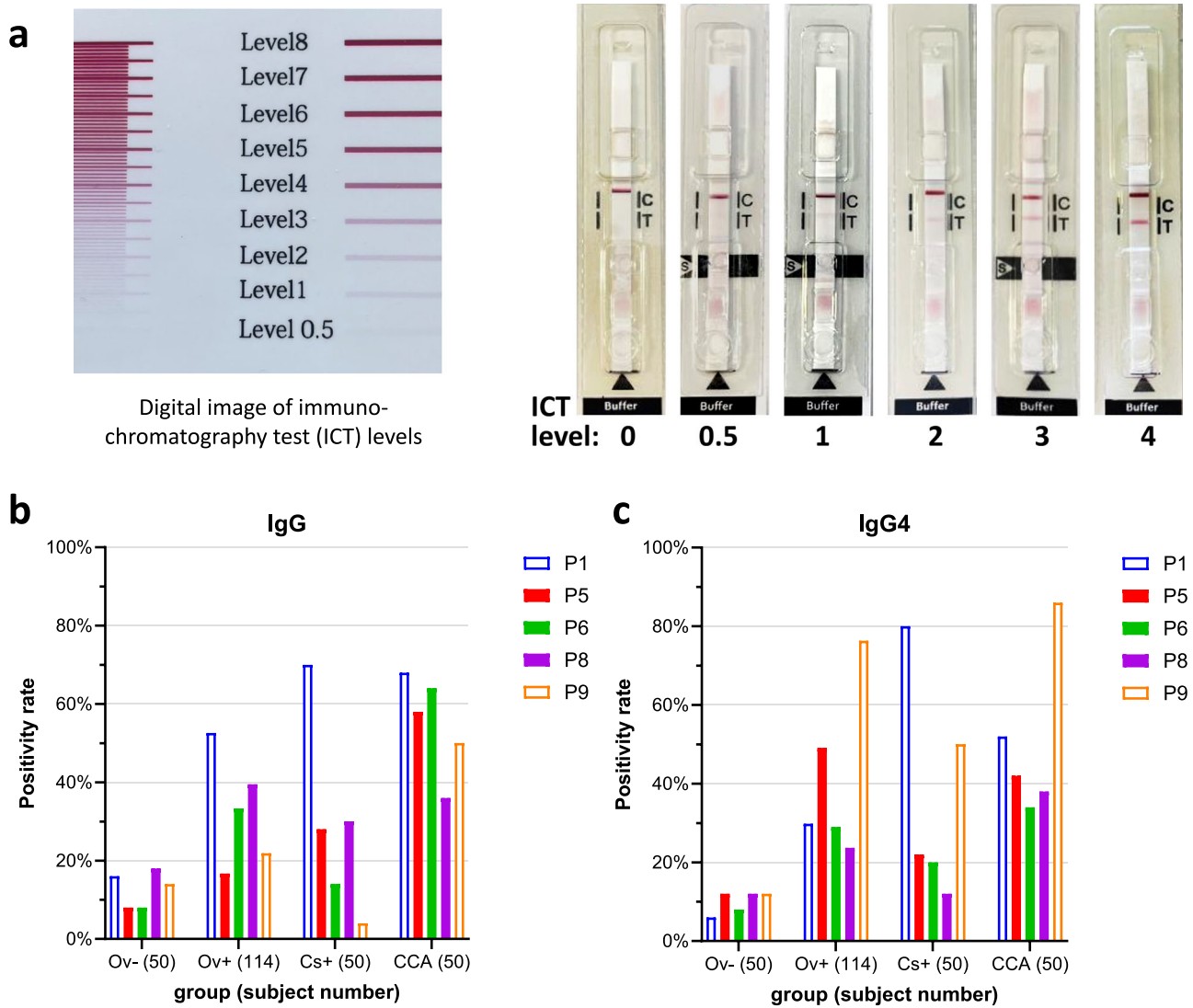

**Fig. 3 | Immunoreactivity of *Opisthorchis viverrini* recombinant proteins printed on immunochromatography tests (ICTs) for the serodiagnosis of opisthorchiasis, clonorchiasis and opisthorchiasis-associated cholangiocarcinoma (CCA).** A positive ICT reaction based upon visual assessment was afforded to a protein if a score of equal to or greater than 0.5–1 was attained; each protein has a defined cut-off as described in Supplementary Table 3. Digital scorecard (left) and ICT strips (right) with scores ranging from 0 to 4 are shown (**a**). C = Control band, T = test

band. The IgG (**b**) and IgG4 (**c**) positivity rates (%) are shown for the 5 candidate antigens with successful ICT formulations. Ov-: 50 healthy uninfected controls from Thailand (endemic and non-endemic); Ov + : 114 *O. viverrini* infected subjects with known EPG (Thailand and Laos); Cs + : 50 *Clonorchis sinensis* infected subjects; CCA: 50 CCA patients where the causative agent was suspected to be *O. viverrini* infection.

We generated a receiver-operator characteristic (ROC) curve to compare the area under the curve (AUC), sensitivity, specificity, and likelihood ratio (of a correct diagnosis) (Fig. 4e). The P9-IgG4 ICT was superior amongst the tests that involved a single recombinant protein (or crude ES products), yielding a sensitivity of 76.3% and specificity of

87.5%. ES-IgG4 ICT was less sensitive at 69.3% but had a higher specificity of 98%. A combination of the P1-IgG and P9-IgG4 ICTs yielded 88.6% sensitivity, superior to either recombinant protein test alone as well as to the ES-IgG4 ICT, although specificity fell to 70.8% when both antigens were included. To compare the efficacy of the various pilot

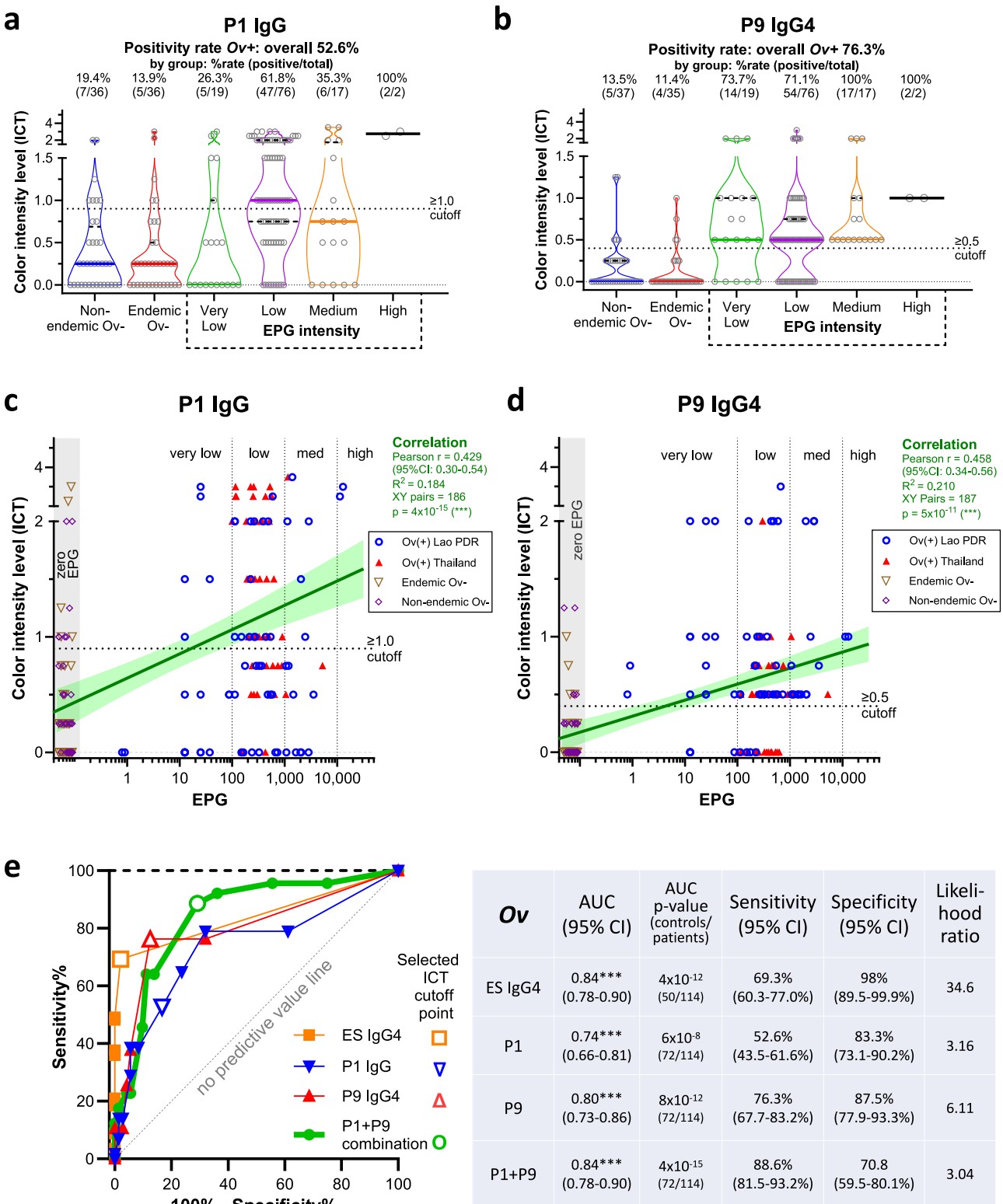

ICTs against FECT, we conducted a kappa test, noting that all subjects were determined to be infected based on detection of eggs in feces, and hence FECT rates were therefore 100%. The best performing tests, P9-IgG4 and P9-IgG4/P1-IgG combination obtained kappa scores (95% CI) of 0.610 and 0.607, respectively, equating to kappa agreement levels of substantial (P9-IgG4) and moderate (P9/P1 combination) (Supplementary Table 1).

Finally, the predictive values of the P1-IgG and P9-IgG4 ICTs were determined using a ROC curve for a cohort of 50 patients with CCA

that was suspected to be caused by chronic *O. viverrini* infection (Fig. 5). The control group consisted of the same endemic and non-endemic sera employed for infection status but, here, they were pooled into an *O. viverrini*-negative group. Thirty-eight of the 50 sera were also available for subsequent testing with a previously developed *O. viverrini* ES-IgG ICT[11], which served as a benchmark for serological diagnosis. Both P1-IgG and P9-IgG4 out-performed ES-IgG ICTs for their respective antibody subclasses, including in combination form (Fig. 5a–c). For example, 75% of cases were positive for at least IgG or

**Fig. 4 | Association between *Opisthorchis viverrini* infection intensity and positivity rate of point-of-care immunochromatographic tests (PoC-ICTs) tests for P1 and P9 antigens.** P1-IgG (**a**), and P9-IgG4 (**b**) ICT reactivity scores of 114 *O. viverrini* infected subjects ranked by eggs per gram of feces (EPG) intensity, and endemic (Thailand) and non-endemic (Australia and Thailand) uninfected controls. Uninfected control totals vary as not all subjects were tested due to sample limitations. The median values are denoted by thick colored lines and quartiles as black dashed lines. EPG intensity was categorized as follows: Very low, 0-100 EPG; Low, 100-1000 EPG; Medium, 1000-10,000 EPG; High, >10,000 EPG. Positivity cut-off is marked with a dotted line. Two-sided Pearson correlation coefficient for P1-IgG-ICT (**c**) and P9-IgG4-ICT (**d**) scores and fecal egg counts: *** = $p < 0.001$. Linear regression shown by dark green line with 95% confident intervals (CI) shown as shaded light green. Positive threshold cutoff marked with dotted horizontal line. Vertical dotted lines mark low/medium/high EPG intensities. EPG values of 0 were given nominal values of 0.05–0.09 to enable plotting on log EPG axis; these values are marked by the grey shaded region. Comparison of P1-IgG, P9-IgG4 and ES-IgG4 ICTs for diagnosing *O. viverrini* infections (**e**) with Receiver-operator characteristic (ROC) curve and output table showing the superiority of the combined P1-IgG and P9-IgG4 immunochromatographic tests irrespective of infection intensity. The diagnostic accuracy of P1, P9 and P1/P9 to detect antibodies in the sera of *O. viverrini* infected subjects was measured by the area under the ROC curve, and the likelihood ratio for a positive diagnosis using P1/P9 was determined.

IgG4 against ES products whereas 92% of patients were positive for at least one of the P1-IgG (AUC 0.84) or P9-IgG4 ICTs. Indeed, P9-IgG4 ICT alone identified 86% of the CCA cases (Fig. 5d). We further analyzed the seroreactivity of CCA patients with the two ICTs based on their CCA grading. Neither ICT was better at differentiating one grade of CCA from another, however there was a trend, although not statistically significant, towards increasing median ICT score with higher stages of cancer (Fig. 5e). We also examined seroreactivity of CCA patients with the two ICTs based upon liver location or CCA type (Supplementary Fig. 5). Intrahepatic (26/51) was the most common location and mass forming was the most common CCA type (23/51). These two categories had elevated seropositivity using the P9-IgG4 ICT - mass forming CCA (22/23, 95.6%) or intrahepatic CCA location (25/26, 96.2%) – compared to overall 86% detection.

## Discussion

NE Thailand has historically reported the world's highest incidence of CCA, at >80 cases per 100,000 population, and opisthorchiasis-associated CCA is thought to kill up to 20,000 people every year in Thailand[2]. Indeed, there seems to be no stronger link that occurs between malignancy and infection with a eukaryotic pathogen than that with CCA and *O. viverrini*.

The possible liver fluke seropositivity in American veterans of the Vietnam War[18] and corresponding increase in CCA in veterans[19] raises the possibility that some U.S. soldiers infected with liver flukes when deployed in East Asia might have developed CCA in later life[18]. Psevdos and colleagues noted the absence of an *O. viverrini* recombinant protein-based test to provide reliable serodiagnosis of current or past infections with the fluke. Similarly, there is interest in development of PoC serological tests to diagnose *O. viverrini* infection throughout Southeast Asia due to the laborious nature of FECT and the need for specifically trained personnel. A field-deployable PoC that could diagnose current/recent infection is therefore urgently required. Moreover, if such a test was capable of diagnosing fluke-associated CCA or identifying infection cases and hence risk of developing CCA, it could save lives by facilitating early and targeted intervention.

FECT remains the gold standard for diagnosis of opisthorchiasis and clonorchiasis[20], despite its inadequacy for diagnosing infections/transmission in elimination and post-elimination surveillance settings. Serodiagnosis is available in Thailand and elsewhere, for example using ELISA[21] and PoC-ICTs that contain crude ES proteins from cultured adult liver flukes[11], but these tests are for research purposes only and are not widely available. Assays that require parasite-derived reagents suffer from availability and reproducibility impediments and cannot be readily standardized across laboratories. To surmount this limitation, we constructed a secretome-scale recombinant proteome microarray and screened it with sera from individuals known to be stool positive for infection to identify proteins with diagnostic potential.

P1 is a member of the peptidase C1 family of cysteine proteases with greatest identity to cathepsin F and was detected in the tegument, ES and EV proteomes (table S3). *O. viverrini* secretes a family of closely related C1 proteases, and P1 shares 94% amino acid sequence identity with *Ov*-CF-1, a protease that is produced in the gut and tegument of the fluke and is detected in cholangiocytes surrounding adult flukes in the bile ducts of infected hamsters[22]. Anti-*Ov*-CF-1 antibodies were detected by ELISA in 50% of FECT-positive *O. viverrini* infected subjects[23], and the *Ov*-CF-1 antigen was detected in the feces of 93% of 30 infected subjects using a chicken IgY antibody raised to the recombinant protein, although the specificity was only 76% due to cross-reactivity with other confirmed helminth infections[24]. *O. viverrini* P1 had a sensitivity of 52.6% for the IgG-ICT and 80.0% for the IgG4 ICT for *C. sinensis* infected subjects. The predictive value was higher for *C. sinensis* infected subjects than it was for *O. viverrini* infected subjects; however, we are hesitant to make quantitative comparisons because the average infection intensities for the *C. sinensis* infected subjects (average 5320 EPG, median 1224) was higher than that for *O. viverrini* infected subjects (average 748 EPG, median 294).

Faced with cross-reactivity and relatively poor sensitivity for P1 as a diagnostic antigen for opisthorchiasis, we explored the predictive value of other antigens. The P9-IgG4-ICT diagnosed 76% of FECT-positive *O. viverrini* cases, while 12% of FECT-negative subjects received a positive diagnosis (5/37 non-endemic subjects and 4/35 endemic subjects). It is feasible that the endemic subjects who were FECT negative had IgG4 to P9 as result of a low intensity infection that was not detected by FECT, or they remained seropositive from an earlier infection, however a similar percentage of non-endemic subjects were also seropositive by ICT, suggesting that these subjects were false positives. P9 is a member of the NADP-dependent isocitrate dehydrogenase (IDH) family and was detected in the tegument, ES and EV proteomes (Supplementary Data 2, Supplementary Fig. 1).

When the findings for P1- and P9-IgG4-ICTs were combined, the sensitivity increased to 89% (Supplementary Fig. 2) but the false positivity rate also increased to 18%. Intriguingly, none of the false positives were IgG4 seropositive for both P1 and P9 antigens; each subject only recognized one or the other antigen. Significant correlations were detected for P1-IgG-ICT responses and infection intensity as well as P9-IgG4-ICT responses and infection intensity. Although there were only two cases with highly elevated egg counts, infection intensity correlated strongly with these two antigen-specific responses (Fig. 4).

The P9-IgG4-ICT proved to highly sensitive for diagnosis of fluke-induced CCA in our cohort, with a predictive value of 86%. This increased to 92% when the P1-IgG-ICT was included, where a positive diagnosis was considered if either or both tests were positive. This is a substantial improvement over the predictive value obtained with IgG- and IgG4-ES ICTs, in both this study and in earlier reports[25]. This finding is notable given that fluke-induced CCA cases often do not remain infected by the point of CCA diagnosis (including when blood was drawn for this study) due to their advanced clinical care which usually involves praziquantel treatment[26]. It is highly likely that most of these CCAs were caused by chronic liver fluke infection given this is the most frequent cancer diagnosed in this region of Thailand where *O. viverrini* is endemic[2,27]. The combination of both P1 and P9 ICTs, or even the P9-IgG4 ICT alone will rapidly identify infected subjects who are at high risk of developing CCA such that anthelmintic chemotherapy can be administered to reduce the likelihood of cancer

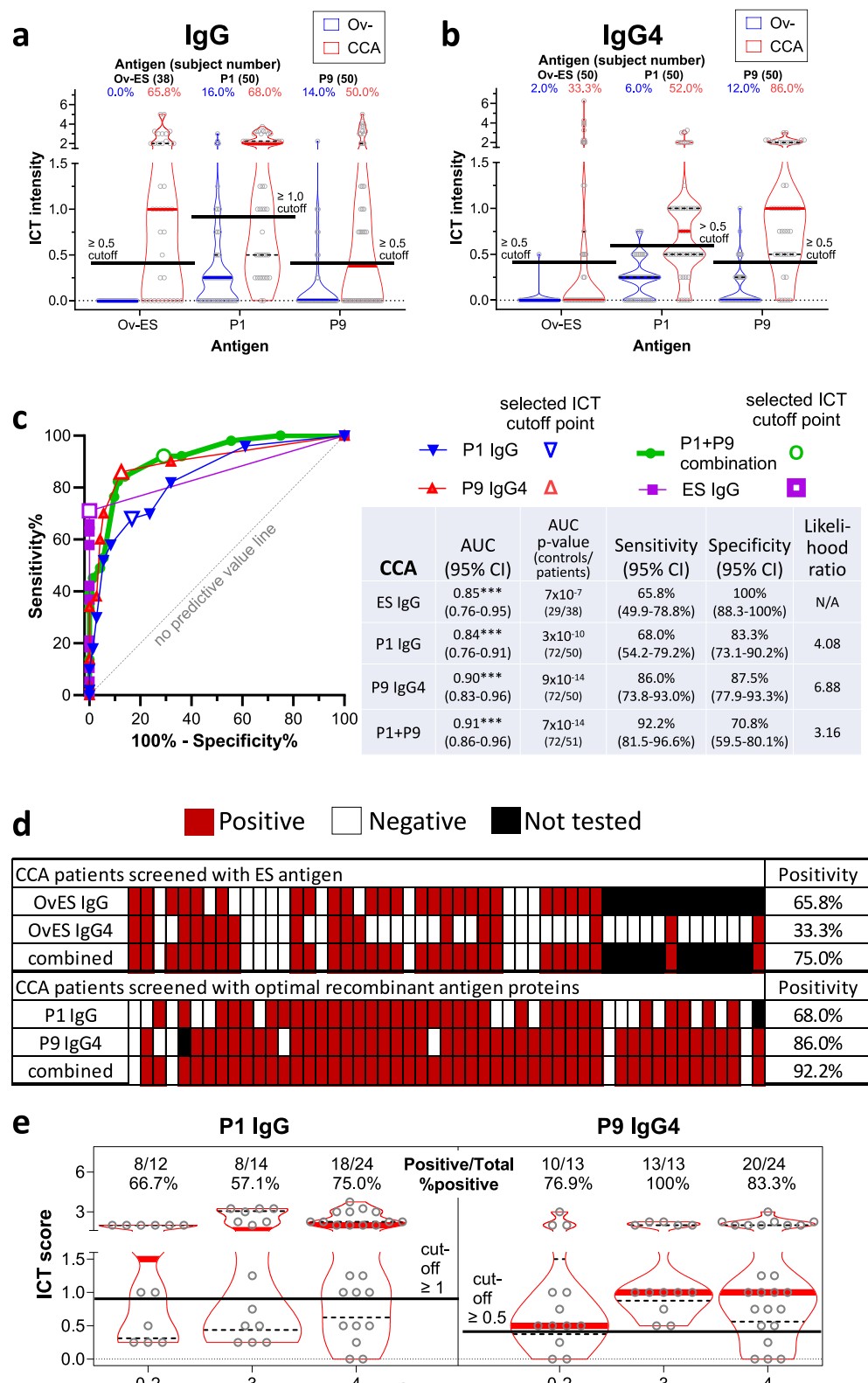

development. The immunopathogenesis of chronic opisthorchiasis is a spectrum of conditions, commencing with changes to the liver that occur before disease sequelae are apparent, including development of asymptomatic advanced periductal fibrosis accompanied by inflammatory cytokine production[26,27]. The P9-IgG4-ICT may add value in identifying asymptomatic subjects who are at elevated risk of developing periductal fibrosis and ultimately CCA.

Future research needs to address antibody decay rates in a treatment-based study to understand how rapidly subjects convert from seropositive to seronegative in the absence of ongoing transmission/exposure. Given that antibodies persist in the host after liver fluke has been eliminated, we posit that the P1/P9 antibody response seen in egg-negative individuals who reside in endemic areas resulted from infection in the previous several months. In the absence of

**Fig. 5 | Positivity rate of point-of-care immunochromatographic tests (PoC-ICTs) tests for P1, P9 and P1/P9 combined antigens for the serodiagnosis of opisthorchiasis-associated cholangiocarcinoma (CCA).** Comparison of PoC-ICT scores for serum IgG (**a**) and IgG4 (**b**) responses of *Opisthorchis viverrini*-associated CCA patients compared to endemic uninfected (Thailand) and non-endemic (Australia and Thailand) serum donors to P1, P9 and *O. viverrini* ES products. The median values are denoted by thick grey lines and quartiles as black dashed lines. Positivity cut-offs are marked with solid black lines. Receiver operating characteristic (ROC) curves and output table of IgG and IgG4 reactivity to ES products, P1, P9 and combined P1/P9 (**c**). The diagnostic accuracy of P1, P9 and P1/P9 to detect antibodies in the sera of CCA patients was measured by the area under the ROC curve, and the likelihood ratio for a positive diagnosis using P1/P9 was determined. The heat map illustrates the data from (**a**) and (**b**), with positive responses denoted by the red and negative responses denoted by white cells (**d**). Black cell = not tested. Diagnostic accuracy was also assessed based on different clinical stages of CCA, stages 0-2, 3 and 4 (**e**). Violin plots show median values as thick red bars with dashed line quartiles. Solid black lines denote positivity cutoffs. Positive samples/total samples and percent positive rates are reported above each plot. Cancer stages 0–2 are grouped together due to low sample numbers for each stage.

corroborative testing, a potential limitation of using antibodies to diagnose infection can be the inability to distinguish past and current infection. Regardless, the utility of antibody-based diagnosis, particularly in monitoring of liver fluke transmission in areas where elimination has been achieved, and praziquantel treatment effect monitoring is taking place should not be underestimated. We are not aware of a published target product profile (TPP) for serodiagnosis of human liver fluke infections. There are clearly, however, key differences in TPP for the two case scenarios we investigated herein. An infection-focused test should be highly sensitive and field-deployable. Moreover, such a test can be used in endemic areas to rapidly screen for liver fluke infection, including where infections may be asymptomatic or mild, and where there are complications that can include hepatobiliary morbidities such as hepatomegaly, cholangitis, cholecystitis, periductal fibrosis and/or gallstones along with a negative stool examination finding for *Opisthorchis* eggs. An antibody-based test cannot distinguish between current and recent past infection due to antibody decay/persistence rates, but antibodies are a useful biomarker for ongoing or recent low-level transmission, particularly in elimination settings[13]. On the other hand, a CCA-focused test should be highly specific and needs to differentiate from unrelated hepatobiliary conditions and malignancies and be integrated with clinical workflows for managing and treating CCA[2]. Tests such as those we now report will significantly enhance surveillance and early diagnosis, ultimately reducing the burden of opisthorchiasis-related complications in endemic regions.

Downstream research might also assess the utility of the test for diagnosing infections with the related liver flukes, namely *C. sinensis* and *O. felineus*. Cross-reactivity of the test during infection with these flukes is not a concern for diagnosing opisthorchiasis because the different species have different snail intermediate hosts and do not geographically overlap. Nonetheless, it is important to understand the potential utility of these tests for diagnosing *C. sinensis* and *O. felineus* infections. The P1-IgG and IgG4 ICTs diagnosed 70% and 80% of *C. sinensis* infected subjects, respectively, and was the antigen with the greatest predictive value for clonorchiasis.

To conclude, our integrated immunomics approach identified informative antigens using a novel protein microarray for the serodiagnosis of opisthorchiasis and opisthorchiasis-associated CCA. Two of the most promising antigens, P1 cysteine protease and P9 isocitrate dehydrogenase, were each incorporated into a field-deployable PoC-ICT to further validate their diagnostic performance. Studies to address the performance of pilot PoC-ICTs using finger prick blood will likely be informative, as was described using the same ICT format for diagnosing human gnathostomiasis by our team (WM[28]). The positive outcome of this investigation represents a truly bench-to-field approach to advance the development of human liver fluke infection and CCA diagnostics.

## Methods
### Ethics statement
Ethics approval for collection of samples from individuals from Thailand and Lao PDR was obtained from the Khon Kaen University Ethics Committee for Human Research (HE611507, HE631300, HE641114, HE641242 and HE664044). All subjects gave written consent for use of their samples specifically for this study or for related studies. For CCA patients, informed consent was obtained from 21 participants, whereas the remaining 30 CCA cases were sourced from leftover tissues where use of the material was permitted for related projects by informed consent. Other serum samples from infected subjects (not CCA patients) were provided under informed consent for non-specific projects and the Khon Kaen University Ethics Committee had waived requirements for project-specific informed consent. Clonorchiasis serum samples were approved by the Medical Ethics Committee of the National Institute of Infectious Diseases, Tokyo, Japan (Nos. 177 and 589). Collection and use of samples from individuals from Australia was approved by the James Cook University Human Research Ethics Committee under approval H8523. A de-identified panel of healthy control sera for protein microarray serology was collected at University of California Irvine General Clinical Research Center (GCRC), now Center for Clinical Research, under an approved protocol HS# 2007-5896. Excretory/secretory products of *O. viverrini* were obtained from flukes passaged through hamsters with approval from the Animal Ethics Committee of Khon Kaen University and according to the Ethics of Animal Experimentation of the National Research Council of Thailand (AEMDKKU 002/2018).

### *O. viverrini* secreted proteomes and microarray construction
The *O. viverrini* secretome, consisting of soluble excretory/secretory products (ES), exosome-like extracellular vesicles (ELVs) and microvesicles (MVs) were analysed by tandem mass spectrometry. A total of 20 μg of *O. viverrini* soluble excretory/secretory (ES) products was dissolved in 50 mM ammonium bicarbonate ($NH_4CO_3$) and 20 mM dithiothreitol (DTT), followed by incubation at 65 °C for 60 min. Alkylation was carried out by adding iodoacetamide (IAM) to a final concentration of 55 mM and incubating the mixture in the dark at room temperature for 40 minutes. Subsequently, a final incubation with 100 mM DTT was conducted at room temperature before adding 1 μg of trypsin and incubating the mixture at 37 °C overnight. Thereafter, peptides were desalted using a Zip-Tip (Millipore Sigma, Burlington, MA, USA) and analysed by mass spectrometry using a Triple TOF 5600+ mass spectrometer (AB SCIEX) with a nano electrospray ion source coupled to a LC-MS/MS on a Shimadzu Prominance Nano HPLC as described[29]. Briefly, 15 μl of digested peptides was injected onto a 50 mm×300 μm C18 trap column (Agilent Technologies, Santa Clara, CA, USA) at a constant flow rate (60 μl/min). Following desalting, the trap column was connected to a 150 mm×100 μm 300SBC18, 3.5 μm nano HPLC analytical column (Agilent Technologies) and peptides were separated using a linear gradient of 2–40% solvent B (90% acetonitrile, 0.1% formic acid in water) over 80 minutes at a flow rate of 500 nL/min. This was followed by a 6-minute wash at 2% solvent B and a steeper gradient increasing from 40% to 80% solvent B over 10 minutes. Solvent B was maintained at 80% for 5 minutes to wash the column and then reduced to 2% for re-equilibration before the next sample injection. The mass spectrometer settings included an ionspray voltage of 2200 V, a declustering potential of 100 V, a curtain gas flow of 25, a nebuliser gas 1 (GS1) of 12, and an interface heater temperature of 150 °C. The mass spectrometer acquired 250 ms full scan TOF-MS

data, followed by twenty 250 ms full scan product ion data in Information Dependent Acquisition (IDA) mode. Full scan TOF-MS data was collected over the mass range of 300–1600, and for product ion ms/ms, the range was 80–1600. Ions detected in the TOF-MS scan that exceeded a threshold of 150 counts and had a charge state of +2 to +5 triggered the acquisition of product ion ms/ms spectra of the 20 most intense ions. Data acquisition was managed using Analyst TF 1.6.1 (AB SCIEX).

Newly generated mass spectrometry data from *O. viverrini* ES (PRIDE project PXD056031) and previously reported mass spectrometry data from *O. viverrini* extracellular vesicles (PRIDE projects PXD020356 and PXD020345), including ELVs and MVs[30,31] were reanalysed using a proteomics database based on newly generated genomic data (GenBank PRJNA230518) appended to the common repository of adventitious proteins (cRAP) contaminants database. Other sequences including T265_16380, T265_07410 and T265_10096, and mucinase Ov-M60[32] were added to generate a final database. Analysis was performed using SearchGUI v 3.3.15[33] using X! Tandem v2015.12.15.2[34], MS-GF+ (v2018.04.09)[35] and Tide[36] using the following parameters: trypsin (2 miss cleavages) as enzyme, carbamidomethylation of C as fixed modification, deamidation of N and Q and oxidation of M as variable modifications, 20 ppm as precursor mass tolerance and 0.2 Da as fragment mass tolerance and precursor charges +2 to +4. Data were further processed with peptideShaker 1.16.40[37] to generate a report file. PSMs, peptides and proteins were validated at a 1.0% False Discovery Rate (FDR) estimated using the decoy hit distribution.

Proteins positively identified with ≥2 peptides in all three samples (ES, ELVs and MVs) were included in the final list of selected candidates (Supplementary Data 1 and 2). Additionally, proteins found in two of the samples and manually curated and selected based on homology with diagnostic candidates in other trematodes were also included. Proteins were searched for transmembrane domains and signal peptides using TMHMM - 2.0. and SignalP v4.0, respectively. Signal peptides were removed from the final sequences. For cloning, ORFs corresponding to proteins >120 kDa were split into ORFs of 1779 bp. The ES products and MV mass spectrometry proteomics data have been deposited in the ProteomeXchange Consortium via the PRIDE partner repository with the dataset identifiers PXD056031 and PXD02035, respectively.

## Study design and cohorts
Participants in the cohorts from *O. viverrini*-endemic areas all agreed to FECT to determine positive vs negative status, although quantified egg counts (eggs per gram of feces; EPG) were not available for all cohorts (Supplementary Table 2). Non-endemic control cases were collected from southern and central regions of Thailand, non-endemic areas for opisthorchiasis, and no parasite material was detected in stool examinations using the concentration method[38]. These populations also had no history of consuming raw fish. The endemic control cases were collected from the northeastern part of Thailand from an opisthorchiasis endemic area, but these subjects were negative for parasite infection by stool examination[38]. These subjects were also interviewed and had no history of consuming raw fish. Blood was collected at the same time as feces. Subjects who were negative for parasite eggs by FECT but were seropositive could be explained as subclinical cases and/or harbouring low intensity infections that were not detected by FECT, or they remained seropositive from an earlier infection. The proteome microarray was screened with 50 sera from healthy uninfected and 50 sera from infected subjects from Northeast (NE) Thailand where cases were considered FECT-positive or -negative but EPG were not available for positive subjects. Additional cohorts of 50 FECT positive cases with known EPG from NE Thailand, 64 FECT positive cases from Loa PDR, and 50 FECT positive individuals with quantified *C. sinensis* egg counts from China were included for screening and validating ICTs. Finally, a cohort

of 50 liver cancer patients from NE Thailand with CCA that was suspected to be caused by *O. viverrini* but for which only two patients remained FECT positive was used to screen the proteome array and ICTs. Clinical diagnosis of CCA and stage of disease was recorded for patients (Supplementary Data 3). ICTs were screened with sera from healthy uninfected (EPG negative) Thai subjects resident in endemic (n = 37) and non-endemic (n = 15) areas. ICTs were also screened with sera from uninfected residents of the USA (50 participants) and Australia (22 participants), regions that are not endemic for these fish-borne flukes.

## Microarray assembly and printing
Open reading frames for all selected proteins (Supplementary Data 2) were codon optimised for expression in *Escherichia coli* and commercially synthesised and cloned in pXI vector (Twist Bioscience, San Francisco, CA, USA). DH5α competent cells were transformed using synthesised cloned genes for plasmid isolation. Proteins encoded by each purified plasmid were expressed in vitro (RTS 100 *E. coli* HY kit – Biotechrabbit, Berlin, Germany) according to the manufacturer's instructions and printed onto eight-pad nitrocellulose-coated AVID glass slides (Grace Biolabs, Bend, OR, USA) with an Omnigrid 100 microarray printer (Genomic Solutions, Ann Arbor, MI, USA). Vector containing no insert was similarly "expressed" and printed (in multiple locations) to serve as a negative control and multiple empty spots were left on each pad to serve as background controls. Purified human IgG and Ig-subclasses (IgG1 and IgG4), anti-human IgG and parasite extracts (*O. viverrini* adult stage ES products[39] and somatic antigen[40]) were also printed as positive controls. Expression quality control was assessed by detection of N-terminal HIS tags as previously described[41].

## Probing of *O. viverrini* protein arrays with human sera
For initial screening of the proteome array, we selected 50 serum samples from each of the three cohorts; (i) *O. viverrini*-FECT positive subjects from Khon Kaen province, Thailand (quantitative EPG data were not available but they were known to be FECT positive); 50 sera from uninfected FECT-negative Thai residents from the same endemic area; 50 sera from healthy US volunteers from a non-endemic site. Serum IgG responses to arrayed antigens were determined by probing with human sera diluted 1:50 in array blocking buffer/10% *E. coli* lysate as previously described[41] with the exception that a biotinylated anti-human IgG, IgG1 or IgG4 (1:200 in array blocking buffer) was used as the secondary/detection antibody followed by streptavidin-Qdot 655 (1:200 in array blocking buffer). Protein array blocking buffer was sourced from GVS North America (Sanford ME, USA).

## Recombinant protein production
The nine most reactive antigens from the proteome array were selected for progression to recombinant expression in *E. coli*. Open reading frames were cloned into the *Nde*I and *Xho*I sites of the pET41a *E. coli* expression vector, such that the vector's N-terminal GST tag was removed, to prevent the detection of non-specific immune responses upon ICT validation of recombinant proteins. Pilot expression assays showed P1-4 to be expressed in soluble form with P5-9 expressed in insoluble form. Expression and purification using immobilised metal ion affinity chromatography (IMAC) of soluble proteins was carried out in *E. coli* BL21(DE3) as described by us elsewhere[13]. Insoluble proteins were expressed as inclusion bodies and purified as described with the addition of 6 M urea creating denaturing conditions. Fractions containing recombinant proteins (as determined by SDS-PAGE) were pooled and concentrated using Amicon Ultra-15 centrifugal devices with a 3 kDa molecular weight cut-off and quantified using the Pierce BCA Protein Assay (ThermoFisher Scientific, Waltham, MA, USA. The final concentration of the proteins was adjusted to 2 mg/ml, and proteins were aliquoted and stored at −80 °C.

## Pilot development of PoC-ICTs

The recombinant proteins were successfully expressed and purified for printing on ICTs at a concentration of 2 mg/ml in PBS: P1 (OON19686.1), and 4 antigens at 2 mg/ml in 6 M urea/PBS: P5 (Ov-M60-2); P6 (OON17288.1_C); P8 (OON14063.1); and P9 (OON23642.1). ICT components for P1, P5, P6, P8 and P9 antigens were laminated in five layers: (1) modified backing card (paper lower cassette), (2) nitrocellulose membrane (Sartorius Stedim Biotech SA, Goettingen, Germany) containing a recombinant antigen line (T-line) and anti-mouse IgG antibody (Lampire Biological Laboratories, Pipersville, PA, USA) line (C-line), (3) conjugated pad of antibody-labeled gold nanoparticle, (4) sample pad (Sigma Millipore) and (5) absorbent pad (Whatman Schleicher & Schuell, Dassel, Germany). The laminate was cut using a guillotine into strips of 0.4 cm in length using a guillotine. The test strip laminate was inserted into a plastic cassette cartridge, and the cassette was closed with a cover (Adtec Inc, Oita, Japan). The nitrocellulose membrane was coated with each of the recombinant proteins at the T-line (at each concentration of 0.8 μg protein/line) and anti-mouse IgG antibody as C-line, using the XYZ3000 Dispensing Platform (BioDot Inc., Irvine, CA, USA), at a flow rate of 0.1 μL/mm. Monoclonal anti-human IgG or IgG4 antibody-labelled gold nanoparticles (Kestrel BioSciences Co., Pathumthani, Thailand) were sprayed onto a glass microfiber filter (GF33; Whatman Schleicher & Schuell, Germany) at a flow rate of 1 μL/mm to produce the conjugated pad.

## ICT optimization and screening

The cassettes were optimized for antibody (IgG and IgG4) detection of P1-, P5-, P6-, P8- and P9- containing ICTs. Briefly, 5 μL of each diluted serum sample in running buffer was applied onto a sample well (S). Next, 60 μL of running buffer was applied into the buffer well (B). ICT results were read at 15 min with a naked eye and judged as positive or negative by reference to a color card, with the cutoff for positivity defined. The appearance of red bands at the T-line and the C-line were judged as positive, whereas only the C-line appeared in negative cases. In the absence of the red band at the C-line, the test was determined to be invalid. Optimal conditions for ICT detection are shown in Supplementary Table 3 and a schematic outlining the methodology is shown in Supplementary Fig. 6. The overall study design is depicted in Fig. 1.

## Reporting summary

Further information on research design is available in the Nature Portfolio Reporting Summary linked to this article.

# Data availability

Proteome datasets generated during and/or analysed during the current study are available in the ProteomeXchange Consortium via the PRIDE partner repository with the dataset identifiers PXD056031 and PXD02035, respectively. All other data are available in the Supplementary files. Source data are provided with this paper.

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

## Acknowledgements

We thank Dr. Masanori Kawanaka, former chief of Department of Parasitology, National Institute of Infectious Diseases, Ministry of Health, Labour and Welfare, Tokyo, Japan for permission to use the clonorchiasis sera. This study was supported by grants from the National Research Council of Thailand (NRCT): High-Potential Research Team Grant Program (Contract no. N42A670561 to WM, PMI, LS, RR, PB), the Office of the Ministry of Higher Education, Science, Research, and Innovation under the Reinventing University 2024 Visiting Professor Program (WM), the Research Program from Research and Graduate studies, Khon Kaen University (KKU) (grant no. RP66-7-001 to WM), and by award R01 CA164719 (TL, MJS, PMI, PLF, PJB, AL) from the National Cancer Institue, National Institues of Health (NIH), USA. AL is supported by an investigator grant from the National Health and Medical Research Council of Australia (NHMRC) (2008450). The contents of this report are solely the responsibility of the authors and do not necessarily represent the official views of the NRCT, NHMRC, KKU or the NIH. Schematics created in "Biorender.com".

## Author contributions

PMI, WM, PJB, PLF and AL designed and planned the study. LS, RR, MJS, RN, PB, JS, BAT, RdA, AJ, WI, YW and VHM collected the data. VL, AK, KP, WS, SS, BS, TL, YW and HY provided samples. LS, RR, MJS, RN, PB, JS, RdA, AJ, WM, WI, VHM, PLF, PJB, AL and PMI analyzed and interpreted the data. MJS, RN, LS and RR prepared the figures. AL and PJB wrote the first draft of the manuscript. All authors reviewed and edited the manuscript and approved the final version of the manuscript for submission.

## Competing interests

The authors declare no competing interests.
