## [Peer Review file · Nature Communications]

Immunomics-guided biomarker discovery for human liver fluke infection and infection-associated cholangiocarcinoma

Corresponding Author: Professor Alex Loukas

Version 0:

Reviewer comments:

Reviewer #1

(Remarks to the Author)

Food-borne trematodiasis pose a serious public health problem, affecting more than 50 million people worldwide. Liver flukes, including *Opisthorchis viverrini*, *Opisthorchis felinus*, and *Clonorchis sinensis*, are not only parasitic infections but also carcinogenic, contributing to bile duct cancer and other related malignancies.

The prevention and control of liver fluke infections in both endemic and non-endemic areas rely on parasite detection. However, traditional detection methods, while considered the gold standard for diagnosis, are labor-intensive, have low sensitivity, and require skilled technicians.

This study presents promising results, demonstrating the potential for a rapid ELISA-based screening method that could be effectively applied for field detection in South Asia.

The study is a valuable contribution to field detection for the parasitic and parasite-related diseases. Compared to existing literature, it introduces an innovative design and a novel biomarker for rapid schistosomiasis screening, making it an original and significant advancement in the field.

The manuscript presents results based on two biomarkers for *O. viverrini* infection and infection-associated CCA, evaluated in a cohort of patients. However, the study's sample size appears to be a limitation, which may affect the persuasiveness of the findings. I believe that increasing the sample size would strengthen the conclusions and enhance the reliability of the results.

In this study, patients with CCA caused by chronic liver fluke infection were enrolled. However, the interpretation of CCA needs to be described in greater detail in the methods section. Were all patients, especially CCAs, verified through other diagnostic examinations? Please clarify this in the manuscript to ensure the accuracy and reliability of the findings.

Reviewer #2

(Remarks to the Author)

The manuscript relates to the identification of excretory-secretory proteins of *O. viverrini* and generating a protein array of selected targets from the excretory-secretory proteins. These protein arrays were screened with infected and un-infected sera to identify potential biomarkers. This is a well written paper and the flow is good. It would have been helpful if the manuscript included line numbers to help with the comments.

Comments:

1. In the introduction, the authors mention that there have been other tests (antigen-based and antibody-based) that were developed. However, I do not see any comparison or discussion of the current proposed biomarkers (P1 and P9) with either of the available assays in the manuscript. Were the same samples tested for antigen by any of the methods? Ideally, a new assay needs to be compared with existing assays. Although the assays cited are antigen-based, and the current manuscript is antibody based, from a public health perspective, it would finally come down to which assay has better sensitivities in detecting infections while maintaining specificity.
2. In the abbreviated methods, the detailed methods are supposedly in 'Appendix 1'. But the supplemental methods are not labeled as such.
3. The methods mention excretory/secretory (ES) and extracellular vesicles (EVs). But the supplemental methods mention

ES, EVs, ELVs and MVs. This needs to be clarified. Based on the supplementary methods, it is not clear as to which datasets are from this study and which ones are from prior studies.

4. The related supplemental figure (S2) shows the VENN diagram with ES, ELVs and MVs. Where do the P1 to P9 fall in the VENN diagram? Along same line, Table S3 does not really need the last column 'Selected for the array' if all of the proteins are being printed. Further, the title of Table S3 needs to be modified as it is not a list of all the 825 proteins detected by MS.

5. The second half of the study design repeats itself in the array probing methodology. This redundancy can be cleaned up.

6. On a minor note, what is the composition of the 'array blocking buffer'? There is no reference or mention of it in the supplemental methods.

7. Fig 1.

a. The ES part needs to be clarified in the text (all through) if it is going to be used to encompass all forms of excretory-secretory proteins.

b. While it says 245 of the 278 proteins were printed on the array, the last line of the results section on the characterization of the secretome reads "... a total of 278 printed proteins or protein fragments". In addition, previously published diagnostic targets of *O. viverrini* and *Clonorchis* spp resulted in a total of 249 proteins that were selected to be printed. This needs to be clarified.

8. In the 'immunomics' section of the results, the authors describe that P3 was recognized exclusively by sera of individuals with *C. sinensis* infection. How do you explain this? I do not see much of a discussion as to why 'thioredoxin' that is almost universally present is picked up by sera from *C. sinensis* infections. Also, if sera from individuals with CCA are also positive for P3, then is it really 'exclusive'?

9. Fig 2.

a. The results section mentions 36-antigens (excluding OvES and OvSo) were the targets of significantly elevated IgG1 or IgG4 of which 20 were targets with significant values for both IgG1 and IgG4. The legend says top 40 protein hits and the order of them is different. I do not see the data for P3, P6 and P7 in 2A and 2B. While this is totally up to the authors, I feel that having all the protein names in the middle with their corresponding IgG1 histograms on the top and IgG4 histograms on the bottom, would be easier to see which of the 20 targets were commonly found to have significant. Or it can be flipped vertically so that the naming is easier to read, with IgG1 and IgG4 data to the left and right.

b. How do you explain the relative lack of signal in IgG4 for the OvES? How much of the protein was printed on the array? Assuming equal amounts of proteins across, what is the relative abundance of the selected proteins in the ES?

c. Was there a correlation between the array signals and ICT data?

10. Pg 8. Why were the analyses moved from IgG1 to total IgG for the ICT? I do not see any discussion pertaining to this switch. Also going back, the abstract says IgG reactive antigens were taken forward for recombinant expression. This needs to be corrected.

11. Is there a TPP of some sort CCA/Ov infections? What would be an ideal situations for the two case scenarios?

12. Pg 11. Why is it that the CCA cases were 'suspected' to be caused by chronic *O. viverrini* infection? Were they not diagnosed as Ov?

13. In the discussion, I don't think it is appropriate to say a 'genome-scale proteome microarray' was constructed. The protein array was specific to the ES.

14. The authors do allude to it, but the discussion on EPG between *C. sinensis* and *O. viverrini* is at best is speculative as the correlations between egg burden and signal weak to modest.

15. Pg 14. The starting lines on the seropositivity endemic control populations. During the screening process, were the individuals not asked if they have been tested before? Was this a totally new region that was previously never studied?

16. Maybe I missed something, but looked like all the samples were egg positive cases. And the discussion on CCA cases being in a 'non-infected' state was confusing for me. Were the blood and stool sampled at different times? What are the kinetics of antibody levels? Do they drop off quickly too?

Reviewer #3

(Remarks to the Author)

The manuscript title, "Immunomics-guided biomarker discovery for human liver fluke infection and infection-associated cholangiocarcinoma" aims to develop a diagnostic tool, specifically an immunochromatography strip, for the detection of *Opisthochis viverrini* infection and *O. viverrini*-associated cholangiocarcinoma.

The manuscript is well-structured and informative, and the experimental design is clearly outlined. However, one important question arises regarding How to interpretation the results? Specifically, it is unclear how to interpret, when it is positive results indicating *Opisthochis viverrini* infection, *Opisthochis viverrini* infection in the past, or *Opisthochis viverrini*-related cholangiocarcinoma. This question should be further discussed in the discussion section.

Version 1:

Reviewer comments:

Reviewer #1

(Remarks to the Author)

This manuscript presents valuable findings in the field of *Opisthochis viverrini* infection and associated cholangiocarcinoma (CCA). The study offers important insights into chronic liver fluke infection, highlighting a research area that deserves greater attention given its status as a neglected tropical disease. The work is well-conducted and makes a significant contribution to the field. I recommend its publication in Nature Communications after addressing minor revisions (if any are required by

other reviewers).

Reviewer #2

(Remarks to the Author)

I think the authors have answered satisfactorily to all the questions/comments. A couple of minor queries.

When the authors say 'routinely finding' anti-IgG1 secondary antibodies as sub-optimal, is it across the board for all such studies? If so, why use it in the first place? Secondly, since the authors increased the sample numbers by utilizing the same samples used for array screening, how do the results correlate between the IgG1 (array) and the ICT (IgG)?

Reviewer #3

(Remarks to the Author)

All of the raised question have been satisfactorily addressed by authors.

Point by point response

Reviewer 1.

1. The manuscript presents results based on two biomarkers for *O. viverrini* infection and infection-associated CCA, evaluated in a cohort of patients. However, the study's sample size appears to be a limitation, which may affect the persuasiveness of the findings. I believe that increasing the sample size would strengthen the conclusions and enhance the reliability of the results.

Response: We emphasised geographic diversity of serum samples from fluke-infected subjects used in the study (Thailand, Laos, China). The reviewer's point is well taken, so we used the two best ICTs to screen another 50 serum samples from *O. viverrini*-infected subjects where diagnosis was noted as egg-positive or egg-negative, and egg counts were not quantified as they were for the subjects we currently report in the manuscript. Nonetheless, we thought this would be a valuable addition to the paper and have included the information as Supplementary Figure 4. Please note, this cohort was one of those used to screen the proteome microarray (where we were not concerned with quantified EPG), but we had not previously used this cohort to screen the ICTs. With this new cohort of subjects, we observed similar sensitivity and specificity of the ICTs to that reported for subjects where egg counts had been quantified.

2. In this study, patients with CCA caused by chronic liver fluke infection were enrolled. However, the interpretation of CCA needs to be described in greater detail in the methods section. Were all patients, especially CCAs, verified through other diagnostic examinations? Please clarify this in the manuscript to ensure the accuracy and reliability of the findings.

Response: The requested information on CCA patient clinical diagnosis is now provided in Supplementary Data 3. In addition, we have added a new panel to Figure 5 (Fig. 5E) that shows the positivity rate for serodiagnosis of CCA at different clinical stages of malignancy, and we have discussed the findings at the end of the Results (page 5).

Reviewer 2.

1. In the introduction, the authors mention that there have been other tests (antigen-based and antibody-based) that were developed. However, I do not see any comparison or discussion of the current proposed biomarkers (P1 and P9) with either of the available assays in the manuscript. Were the same samples tested for antigen by any of the methods? Ideally, a new assay needs to be compared with existing assays. Although the assays cited are antigen-based, and the current manuscript is antibody based, from a public health perspective, it would finally come down to which assay has better sensitivities in detecting infections while maintaining specificity.

Response: The antigen detection RDT for opisthorchiasis relies on detection of antigen in urine of infected subjects. Antigen is detected using a monoclonal antibody raised against an undefined antigen, probably glycan in nature based on its Western blot profile. We did not collect urine from our subjects due to ethical constraints. Moreover, the urine antigen test is not widely used now. The gold standard for diagnosing *O. viverrini* infection is the formalin ether concentration technique (FECT). We therefore benchmarked our new antibody ICTs against FECT, as well as an antibody ICT using crude fluke excretory/secretory (ES) products. While we note percentage positivity comparisons in numerous places throughout the

ms, we had not conducted a kappa test to determine the extent of agreement between the two tests. We have now conducted a kappa test comparing the predictive value of our tests with FECT (new Supplementary Table 3) and we have modified the Results to compare positivity rates of our antibody tests with the urine antigen RDT (lines 234-238, page 5).

2. In the abbreviated methods, the detailed methods are supposedly in 'Appendix 1'. But the supplemental methods are not labeled as such.

Response: We apologise for the oversight; we have moved the supplementary methods into the main body of the text such that all the methods are now in the main manuscript.

3. The methods mention excretory/secretory (ES) and extracellular vesicles (EVs). But the supplemental methods mention ES, EVs, ELVs and MVs. This needs to be clarified. Based on the supplementary methods, it is not clear as to which datasets are from this study and which ones are from prior studies.

Response: We have revised the Methods to ensure consistency and make it clear which datasets are new, and which are historical – see section starting at line 471.

4. The related supplemental figure (S2) shows the VENN diagram with ES, ELVs and MVs. Where do the P1 to P9 fall in the VENN diagram? Along same line, Table S3 does not really need the last column 'Selected for the array' if all of the proteins are being printed. Further, the title of Table S3 needs to be modified as it is not a list of all the 825 proteins detected by MS.

Response: As per the journal formatting instructions, we have revised the supplementary information such that tables longer than one A4 page are now referred to as Supplemental Data rather than Supplemental Tables. Supplementary Table 3 in the previous version of the ms is now Supplementary Data 1, and it shows the presence in the three different fluke secreted extracts (ES, ELVs and MVs) of the different proteins. Supplementary Data 2 shows the proteins selected for printing on the array and provides their sequences (listed by accession number). In the ms (page 11, lines 615-617) we had noted the accession numbers for P1 and P9, and these accession numbers can be searched in Supplementary Data 1 and 2 to readily identify the proteins. Nonetheless, to make it easier for readers to access this information, we have revised the Results section to describe that P1 and P9 were detected in all three fluke secretome extracts – ES, ELVs and MVs (lines 215-217). We have also removed the "selected for array" column in Supplemental Data 1 as requested and replaced it with the P1-9 protein codes. We also highlighted P1-9 in the Venn diagram (Supplementary Figure 2).

5. The second half of the study design repeats itself in the array probing methodology. This redundancy can be cleaned up.

Response: We acknowledge the reviewer's concern here but point out that the latter part of the Results section focuses on serodiagnosis of CCA as opposed to the first part of the Results focusing on serodiagnosis of fluke infection. These are two very different objectives that we chose to consider separately to avoid any confusion.

6. On a minor note, what is the composition of the 'array blocking buffer'? There is no reference or mention of it in the supplemental methods.

Response: The blocking buffer is a commercial product of undisclosed composition. We have modified the relevant section in the Methods on page 10 (lines 586-587) to provide the details of the product and manufacturer.

7. Fig 1. a. The ES part needs to be clarified in the text (all through) if it is going to be used to encompass all forms of excretory-secretory proteins.

Response: Thanks for pointing this out. We have rectified the confusing nomenclature as follows: the entire secreted complement (which contains soluble and vesicular proteins) is now referred to as “secretome”; the soluble proteins are “Excretory/Secretory, or ES; exosome like vesicles are ELVs; microvesicles are MVs. b. While it says 245 of the 278 proteins were printed on the array, the last line of the results section on the characterization of the secretome reads “.. a total of 278 printed proteins or protein fragments”. In addition, previously published diagnostic targets of *O. viverrini* and *Clonorchis* spp resulted in a total of 249 proteins that were selected to be printed. This needs to be clarified.

Response: We selected 245 proteins for *in vitro* transcription/translation and printing on the array (as per Supplementary Data 1 and 2). We also included several control proteins that had been expressed in *E. coli* and purified, including Ov-MUC60-1, Ov-MUC60-2, Cs-glutathione transferase omega-1 and Cs-glutathione transferase omega-2. This totals 249 proteins. Some proteins were too large for *in vitro* transcription-translation and were divided into two ORFs. We have revised Figure 1 and its legend to explain this more carefully.

8. In the ‘immunomics’ section of the results, the authors describe that P3 was recognized exclusively by sera of individuals with *C. sinensis* infection. How do you explain this? I do not see much of a discussion as to why ‘thioredoxin’ that is almost universally present is picked up by sera from *C. sinensis* infections. Also, if sera from individuals with CCA are also positive for P3, then is it really ‘exclusive’?

Response: this is a good point, and our wording was poorly chosen. We believe that P3 is the target of an antibody response in *O. viverrini* infected subjects, but the differences in mean IgG1 and IgG4 levels between infected and uninfected subjects did not reach significance. This is likely because most of our *O. viverrini* infected subjects had low or medium intensity infections as opposed to the majority of *C. sinensis* infected subjects having high intensity infection based on FECT. Our CCA patients were mostly FECT negative by the time they were diagnosed with advanced CCA but given that CCA is likely the result of years of chronic (heavy) infection, this is not unexpected. When transitioned to the ICT format, P3 did not perform well compared to P1 and P9. This could have been due to the refolded recombinant protein produced in *E. coli* presenting different epitopes to the *in vitro* transcription/translation product used to print the proteome arrays, or the different substrate and reagents used in the different assays. We have modified the text in the Results (page 4, lines 176-189) to explain this finding.

9. Fig 2a. The results section mentions 36-antigens (excluding OvES and OvSo) were the targets of significantly elevated IgG1 or IgG4 of which 20 were targets with significant values for both IgG1 and IgG4. The legend says top 40 protein hits and the order of them is different. I do not see the data for P3, P6 and P7 in 2A and 2B. While this is totally up to the authors, I feel that having all the protein names in the middle with their corresponding IgG1 histograms on the top and IgG4 histograms on the bottom, would be easier to see which of the 20 targets were commonly found to have significant. Or it can be flipped vertically so that the naming is easier to read, with IgG1 and IgG4 data to the left and right.

Response: we like this suggestion by the reviewer and have redrawn Figure 2A/B to present the data as recommended. Panels A and B are now merged into a single panel A. Proteins were ranked based on IgG4 reactivity P values.

Fig 2b. How do you explain the relative lack of signal in IgG4 for the OvES? How much of the protein was printed on the array? Assuming equal amounts of proteins across, what is the relative abundance of the selected proteins in the ES?

Response: This is possibly due to a small amount of ES being printed on the array and under-representation of IgG4 reactive proteins in this complex mixture. It might also be explained by the relatively light-to-moderate infection intensity of our *O. viverrini* infected subjects, hence the greater reaction with IgG1. This weak IgG4 reactivity to ES products was also seen with PoC ICTs containing crude ES products (Figure 5).

Fig 2c. Was there a correlation between the array signals and ICT data?

Response: we did not detect significant correlations between array and ICT data. The most logical explanation for this is the different matrices used in protein microarray and ICTs, and the different buffer optimisation approaches employed. Many different buffering conditions were assessed in the optimisation process for the ICTs, and some proteins that were reactive on the arrays did not retain equally robust reactivity in ICT format.

10. Pg 8. Why were the analyses moved from IgG1 to total IgG for the ICT? I do not see any discussion pertaining to this switch. Also going back, the abstract says IgG reactive antigens were taken forward for recombinant expression. This needs to be corrected.

Response: When optimising the conditions for printing ICTs, we routinely experienced higher background levels with non-endemic control sera for IgG1 compared to IgG. We have added a statement to this effect in the Results section at the bottom of page 4 (lines 187-189).

11. Is there a TPP of some sort CCA/Ov infections? What would be an ideal situations for the two case scenarios?

Response: We are not aware (from literature searches and discussion with KoLs) of a TPP for serodiagnosis of human liver fluke infections. There are clearly, however, key differences in TPP for the two case scenarios. An infection-focused test should be highly sensitive and field-deployable, while a CCA-specific test should be highly specific (e.g., Needs to differentiate from unrelated hepatobiliary conditions and malignancies) and integrated with clinical workflows for managing and treating cholangiocarcinoma. Such assays will significantly enhance surveillance and early diagnosis, ultimately reducing the burden of opisthorchiasis-related complications in endemic regions. We have expanded the Discussion (lines 411-432) to address this concern.

12. Pg 11. Why is it that the CCA cases were 'suspected' to be caused by chronic *O. viverrini* infection? Were they not diagnosed as Ov?

Response: One can never be certain (with humans) of the cause of cancer. These subjects resided in *O. viverrini* endemic areas, ate raw fish, and some at least had a history of positive FECT diagnosis for *O. viverrini* infection. At the point of clinical diagnosis, many CCA patients no longer harbour flukes, probably due to the inhospitable environment in the cancerous liver. Moreover, CCA is relatively rare in areas where liver flukes are not endemic, and numbers are dwarfed by hepatocellular carcinoma. In unpublished work, we have shown that patients with HCC from fluke non-endemic sites do not have anti-P1 or -P9 antibodies.

13. In the discussion, I don't think it is appropriate to say a 'genome-scale proteome microarray' was constructed. The protein array was specific to the ES.

Response: We have revised the text from "genome" to "secretome" on page 6, line 346.

14. The authors do allude to it, but the discussion on EPG between *C. sinensis* and *O. viverrini* is at best is speculative as the correlations between egg burden and signal weak to modest.

Response: We agree with the reviewer but feel that the Discussion is a suitable place for at least some level of speculation. We noted in the original text that “we are hesitant to make quantitative comparisons because the average infection intensities for the *C. sinensis* infected subjects (average 5,320 EPG, median 1,224) was higher than that for *O. viverrini* infected subjects (average 748 EPG, median 294).”

15. Pg 14. The starting lines on the seropositivity endemic control populations. During the screening process, were the individuals not asked if they have been tested before? Was this a totally new region that was previously never studied?

Response: Non-endemic control cases were collected from southern and central regions of Thailand, non-endemic areas for opisthorchiasis, and no parasite material was detected in stool examinations using the concentration method. These populations also had no history of consuming raw fish. The endemic control cases were collected from the northeastern part of Thailand from an opisthorchiasis endemic area, but these subjects were negative for parasite infection by stool examination (Elkins et al 1986 – new reference 38). These subjects were also interviewed and had no history of consuming raw fish. Blood was collected at the same time as feces. Subjects who were negative for parasite eggs by fecal examination (FECT) but were seropositive could be explained as subclinical cases and/or harbouring low intensity infections that were not detected by FECT, or they remained seropositive from an earlier infection. We have revised the Methods section to include this information at the top of page 10, lines 543-551.

16. Maybe I missed something, but looked like all the samples were egg positive cases. And the discussion on CCA cases being in a ‘non-infected’ state was confusing for me. Were the blood and stool sampled at different times? What are the kinetics of antibody levels? Do they drop off quickly too?

Response: For the serodiagnosis of infection component of the study, all “infected” subjects were diagnosed with the gold standard FECT. Endemic and non-endemic controls were FECT-negative. For the serodiagnosis of CCA cases, blood and stool samples were collected at the same time at the hospital. Studies on antibody kinetics were not performed. Diagnosis of CCA cases was based on gross and histopathological examinations. All cases were located in opisthorchiasis endemic areas and patients all confirmed eating raw cyprinoid fish (*O. viverrini* intermediate host). Two CCA cases were positive for *O. viverrini* eggs by stool concentration (Elkins et al 1986, ref 38). Normally, CCA associated opisthorchiasis is characterized by an absence of *O. viverrini* egg in stool samples, possibly due to chronic opisthorchiasis and associated chronic biliary tract inflammation caused by choledocholithiasis, cholelithiasis, or primary sclerosing cholangitis and bile duct obstruction. We have now included a new Supplementary Data 3 with the stage of CCA diagnosis.

Reviewer 3.

1.one important question arises regarding How to interpretation the results? Specifically, it is unclear how to interpret, when it is positive results indicating Opisthochis viverrini infection, Opisthochis viverrini infection in the past, or Opisthochis viverrini-related cholangiocarcinoma. This question should be further discussed in the discussion section.

Response: We have provided a new section in the Discussion (lines 411-432) focusing on Target Product Profile which addresses the reviewer’s concern.

Point by point response

Reviewers 1 and 3 did not have any remaining questions.

Reviewer 2.

1. When the authors say 'routinely finding' anti-IgG1 secondary antibodies as sub-optimal, is it across the board for all such studies? If so, why use it in the first place?

Response: We were referring to this particular study and noted high levels of background with this particular secondary antiserum. Given the quality and informative nature of the other isotypes, we proceeded without specifically looking at IgG1.

2. Since the authors increased the sample numbers by utilizing the same samples used for array screening, how do the results correlate between the IgG1 (array) and the ICT (IgG).

Response: We did not detect a significant correlation between IgG1 array and IgG ICT data. We did however detect a significant correlation between IgG4 array and ICT results.